**Ozone and Aerosol Optical Depth Retrievals Using the Ultraviolet Multi-Filter Rotating**
**Shadow-band Radiometer**
Joseph Michalsky[1] and Glen McConville[1, 2]
[1]Global Monitoring Laboratory, National Oceanic and Atmospheric Administration,
325 Broadway, Boulder, Colorado 80305 USA
[2]Cooperative Institute for Research in Environmental Sciences, University of Colorado,
216 UCB, Boulder, Colorado 80309 USA
*Correspondence to*: Joseph Michalsky (joseph.michalsky@noaa.gov)

**Abstract:** The ultraviolet multi-filter rotating shadowband radiometer (UV-MFRSR) is a seven-
channel radiometer with narrowband filters centered between wavelengths 300 and 368 nm. Four
of the middle wavelengths in this device are near those used in the Dobson spectrometer to
retrieve ozone column abundance. In this paper measurements from Mauna Loa Observatory
(MLO) were used, first, to calibrate the instrument using the Langley plot method, and,
subsequently, to derive column ozone and aerosol optical depths. The ozone derived from the
UV-MFRSR was compared to the ozone measured by a Dobson spectrophotometer that operates
daily at the MLO resulting in column values within about 1 DU on average for 43 days in 2018.
The aerosol optical depth (AOD) retrievals are more challenging. Generally, the AOD increases
with wavelength between 305 and 332 nm; not what is expected given the typical AOD
wavelength dependence at visible wavelengths. An example of this behavior is discussed, and
research by others is cited that indicates similar behavior at these wavelengths, at least for the
low aerosol optical depth conditions encountered at high altitude sites.
**Ozone Retrieval Introduction**
Most historical network measurements of column ozone from the surface used Dobson or
Brewer spectrometers, and these continue as the predominant ozone measurement instruments
today. Brief explanations of these two devices and comparisons of concurrent and collocated
measurements of total column ozone are given in Staehelin et al. (2003). Gao et al. (2001)
demonstrated that ozone could be retrieved using the ultraviolet multi-filter rotating shadow-
band radiometer (UV-MFRSR), which agreed with those values retrieved from either collocated
Dobson and/or Brewer spectrophotometers to within 1-2%.
The wavelengths used for ozone retrievals in the UV-MFRSR more closely match wavelengths
in the Dobson rather than the Brewer spectrophotometer. Typically, ozone retrieved from the
Dobson uses the AD wavelength pairs 'A' 305.5/325.4 and 'D' 317.6/339.8. Since there is no
filter near 339.8 nm, the UV-MFRSR uses filters near the 'A' pair and the Dobson 'C' pair
311.5/332.4. The filters in the UV-MFRSR that are used for ozone measurements are nominally
the 305/325 nm pair and the 311/332 nm pair with carefully measured profiles of these filters
used for actual retrievals. Normalized filter profiles for UV-MFRSR 453 are shown in Figure 1.

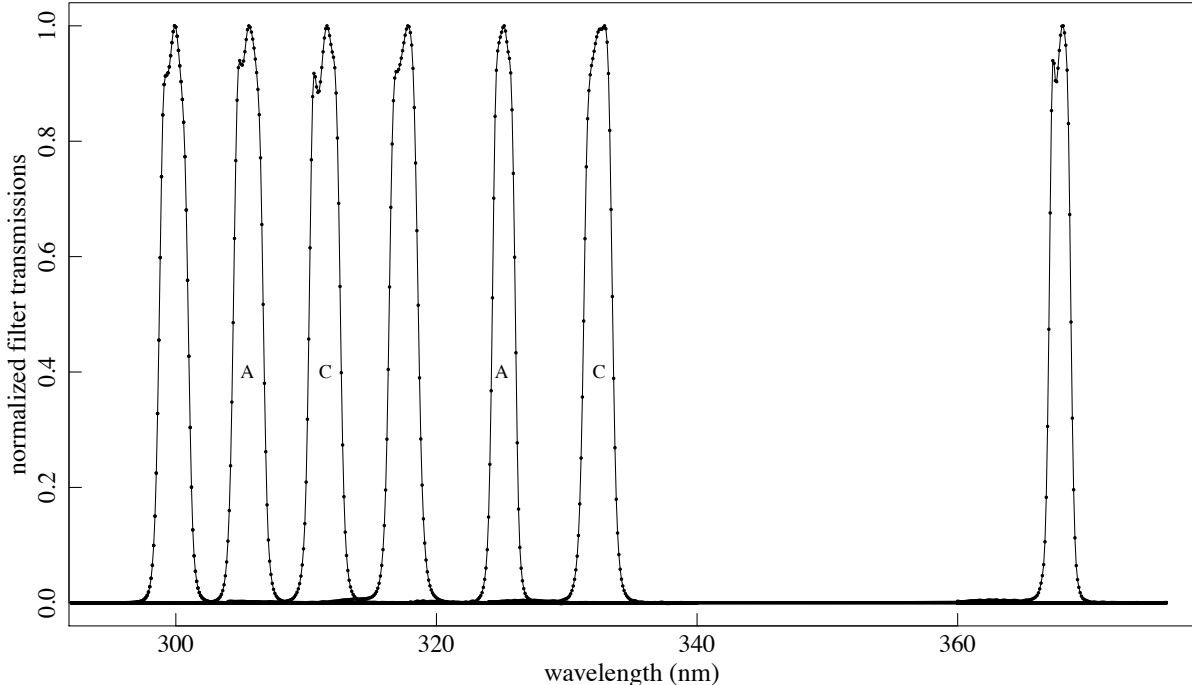

**Filter Profiles for UV−MFRSR 453**

**Figure 1. Normalized filter profiles of UV-MFRSR 453 used in this study. The wavelength-dependent ozone**
**absorption function and Rayleigh scattering function were convolved with these profiles to produce effective**
**absorption and scattering corrections. 'A' and 'C' pairs used for ozone retrievals are noted. Central**
**wavelength/full width at half maximum (nm): 299.9/2.2, 305.6/2.3, 311.4/2.4, 317.5/2.3, 325.1/1.8, 332.4/2.2,**
**367.8/1.7.**

The basic procedure for ozone retrievals consists of measuring extinction at two wavelengths
with one chosen to be more strongly attenuated than the other in the Hartley-Huggins ultraviolet
bands. The basic extinction equation can be written

$$I(\lambda) = I_0(\lambda) \cdot \exp\left[-\tau_{ray}(\lambda)m_{ray}(\lambda)\left(P/P_o\right) - \tau_{oz}(\lambda)m_{oz}(\lambda) - \tau_{aer}(\lambda)m_{aer}(\lambda)\right] \quad (1)$$

or, equivalently,

$$V(\lambda) = V_0(\lambda) \cdot \exp\left[-\tau_{ray}(\lambda)m_{ray}(\lambda)\left(P/P_o\right) - \tau_{oz}(\lambda)m_{oz}(\lambda) - \tau_{aer}(\lambda)m_{aer}(\lambda)\right] \quad (2)$$

since the ratios $I/I_o$ and $V/V_o$ are equal.

In these equations:

$I(\lambda)$ = spectral irradiance measured by the instrument at the surface
$I_0(\lambda)$ = spectral irradiance measured by the instrument at the top of the atmosphere
$V(\lambda)$ = signal (voltage) measured by the instrument at the surface

$V_0(\lambda)$ = signal (voltage) measured by the instrument at the top of the atmosphere
$\tau$'s = optical depths for Rayleigh scattering (ray), ozone (oz), and aerosol (aer)
P, P$_o$ = atmospheric pressure at the measurement site and at sea level, respectively
m's = airmasses for Rayleigh, ozone, and aerosol relative to a vertical path; they differ
slightly because each has a different distribution with altitude in the atmosphere. The
Rayleigh and ozone air masses were calculated using Bodhaine et al., (1999) and
Komhyr and Evans (2008), respectively.

If we write ozone optical depth as $\tau_{oz} = \alpha_{oz} \cdot \eta_{oz}$, where $\alpha_{oz}$ is the ozone absorption coefficient
and $\eta_{oz}$ is the abundance of ozone, we can solve for $\eta_{oz}$ by rearranging terms in two versions of
eqn. (2) representing the two wavelengths in the pair (the longer wavelength is indicated by
primes). Therefore, dropping the explicit $\lambda$ dependence for clarity, we get for ozone abundance

$$\eta_{oz} = \frac{N - (\tau_{ray} - \tau'_{ray})m_{ray}(P/P_o) - (\tau_{aer} - \tau'_{aer})m_{aer}}{(\alpha_{oz} - \alpha'_{oz})m_{oz}}, \tag{3}$$

where N is defined as

$$N = \ln\left(V_o/V'_o\right) - \ln\left(V/V'\right). \tag{}$$

Since all of the parameters of eqn. (3) are known or can be calculated, one could solve for $\eta_{oz}$ if
the term $(\tau_{aer} - \tau'_{aer})$, i.e., the aerosol optical depths at the two wavelengths were known. To
curtail this requirement, the 'A' and 'C' wavelength pairs are used, and the assumption is made
that since the wavelength separation of each pair is nearly the same and the wavelength
dependence over this small wavelength region is expected to be nearly linear, subtraction of eqn.
(3) applied to each pair will come very close to eliminating the aerosol terms because subtraction
of aerosol terms should be near zero if these assumptions hold. The resulting equation used to
calculate ozone is

$$\eta_{oz} = \frac{N_1 - N_2 - [(\tau_{ray} - \tau'_{ray})_1 - (\tau_{ray} - \tau'_{ray})_2]m_{ray}(P/P_o)}{[(\alpha_{oz} - \alpha'_{oz})_1 - (\alpha_{oz} - \alpha'_{oz})_2]m_{oz}}, \tag{4}$$

where

$$N_1 = \ln\left(V_{o,305}/V'_{o,325}\right) - \ln\left(V_{305}/V'_{325}\right),$$

and

$$N_2 = \ln\left(V_{o,311}/V'_{o,332}\right) - \ln\left(V_{311}/V'_{332}\right).$$


**Calibration and Ozone Measurement Comparisons**

The Langley calibration of the UV-MFRSR was performed at NOAA's Mauna Loa Observatory (Latitude = 19.5362°N; Longitude = 155.5763°W; 3397 m). The height of the observatory often allows measurements to be made in clean, free-tropospheric air above the marine boundary layer, especially in the morning hours.

UV-MFRSR data were obtained on 242 days in 2018 beginning on 14 February and ending on 15 October. There were 139 successful Langleys during this period that produced estimated Vo's with only 27 of these during the afternoon hours. Looking at the retrieved Vo's as a function of time there is a hint of a decrease, but not one filter indicates a statistically significant decline, therefore, averages of Vo's over the entire period are used in the ozone and aerosol retrievals.

The process used to choose acceptable Langleys (Michalsky et al., 2001) eliminates Langleys that are influenced by large changes in ozone during a Langley plot. Further, rarely did the standard deviation of the ozone sampled change by more than 5 DU during a morning or afternoon when Langleys plots are sampled. This small change is typical for this low latitude.

Ozone is a standard measurement at NOAA's Mauna Loa Observatory and has been made with near continuous sampling since 1963. The Dobson spectrophotometer there makes AD paired measurements to determine ozone using absorption coefficients measured by Bass and Paur (1985). No estimate of the ozone column below the observatory, which could be on the order of 5% of the column total at sea level, is made. Therefore, the column measurements made using the UV-MFRSR can be directly compared to the Dobson column measurements if one uses the Bass and Paur (1985) absorption cross-sections for the UV-MFRSR channels.

Since the Dobson generally uses the AD pair for the total column ozone calculation, we investigated the difference between AC and AD Dobson retrievals on two clear days at Mauna Loa that were used for Langley calibrations of the Dobson thus giving us more than the operational 1000, 1200, and 1400 local time ozone measurements. It is important to assess any differences since the UV-MFRSR uses wavelengths close to the AC pair for its ozone retrievals. Figure 2 illustrates the difference between Dobson measurements with the two different

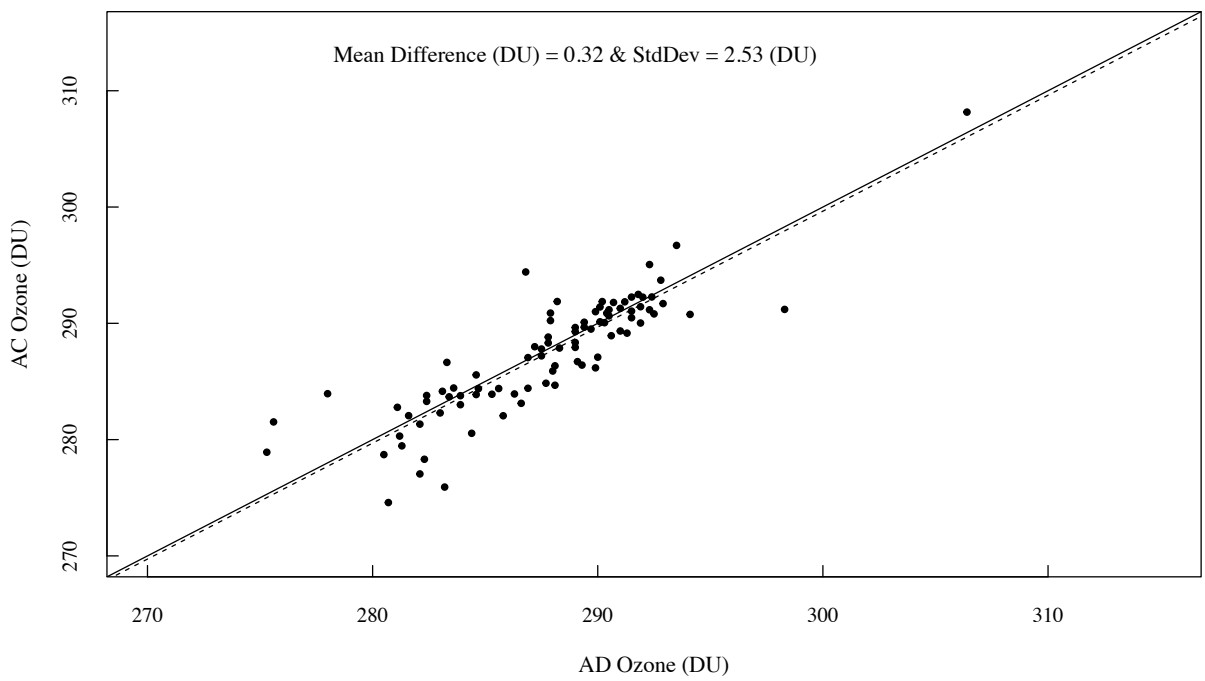

**Figure 2. Plot of ozone measured by a Dobson unit at Mauna Loa Observatory retrieved using the Dobson**
**AC pair versus the Dobson AD pair. Solid diagonal line is 1:1 line and dashed line is linear least-squares fit.**
**The mean difference and standard deviation of the samples are given on the plot.**
wavelength pairs. The mean difference in retrieved ozone for the 90 points compared in the plot
is less than 0.5 DU and the standard deviation among the 90 samples is close to 2.5 DU.
Therefore, using the AC pair of the UV-MFRSR for ozone retrievals and comparing to AD-pair
Dobson ozone should be acceptable.
Figure 3 is a plot of the ozone time series retrieved from the Dobson AD pair and the UV-
MFRSR AC pair for the 2018 data that were matched by day of year. In the case of the Dobson,
one measurement is chosen from the three daily measurements made at 1000, 1200 and 1400

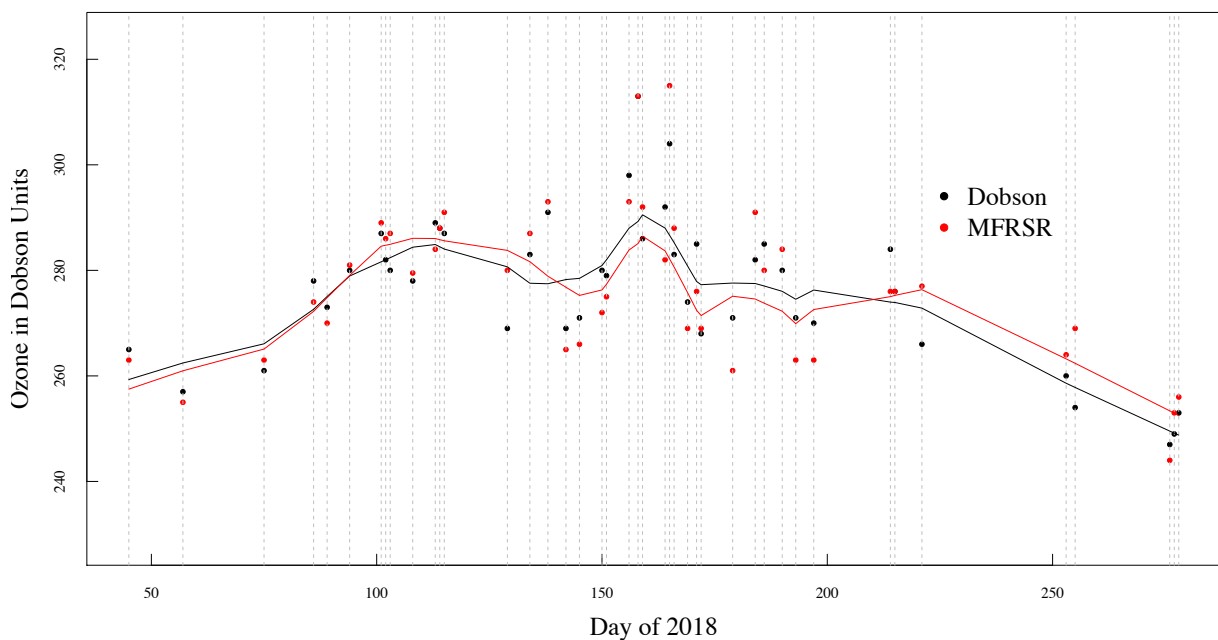

**Figure 3. Time series plot of Mauna Loa Observatory for 43 days of retrieved ozone for 2018 using the**
**Dobson spectrophotometer (black dots) and the UV-MFRSR (red dots). The lines are lowess fits using 0.25 of**
**the points for the lowess fit at each point. The Dobson uses one of three measured points for the daily value,**
**and the UV-MFRSR uses the median of all 20-second, clear-sun data for air masses less than three.**


local standard time. Only direct sun measurements made with the Dobson are used for this
comparison. For the UV-MFRSR data, which is sampled every 20 seconds, a median value of all
points, which are made at less than three air masses and that pass cloud-screening (Michalsky et
al., 2010), is used. Since measurements from the two instruments are made differently and no
attempt to make them coincident, except for occurring on the same day, there is no expectation
of perfect agreement given any diurnal variability. The average difference over the 43-day
sample is about 0.10 Dobson units. The lowess fits to the two data sets track each other rather
closely matching dips and peaks throughout the measurement period.


**Sources of Ozone Uncertainty**

Uncertainties in using a UV-MFRSR for ozone retrievals were discussed thoroughly by Gao et
al. (2001).  In this paper only data taken at less than three air masses (about 71° solar-zenith
angle were used because (1) air mass determination is less certain at higher solar-zenith angles
and the cosine response correction for the UV-MFRSR is larger and more difficult to pinpoint
and, therefore, more uncertain. The extraterrestrial responses for the four filters used to retrieve
ozone were averages for the 242-day period in 2018 as stated earlier. The uncertainties in
extraterrestrial responses were between  0.2% and 0.3%. The ozone absorption coefficients were
those measured by Bass and Paur (1985) adjusted for mid-latitude seasonal variations.  The
effective ozone absorption coefficients were determined by convolving each of the filter profiles
with the wavelength dependent Bass and Paur (1985) ozone absorption coefficients. Similarly,
effective Rayleigh scattering optical depths were determined in the same manner. The effective
Rayleigh optical depths were pressure corrected using on-site measurements of atmospheric
pressure.
Always a major concern when working in the ultraviolet is light from outside the band passes
contributing to the measured signal. Si-C (silicon carbide) is the detector for the 300 nm and 305
nm filters.  GaP (gallium phosphide) is used as the detector in the five longest wavelength filters.
To measure the extent of the possible long-wavelength leakage, we used a Schott glass OG530
placed over the entrance optic being careful to block light paths from the edges that might reach
the entrance diffuser optic. The transmission below 460 nm is 0.00001, therefore no light should
reach the detectors with the OG530 completely covering the entrance optic. If higher orders of
light from the interference filters would reach the detectors, they would begin to be a problem
around 600 nm for the 300-nm filter and at longer wavelengths for the other six filters. The
nighttime dark readings and 530 Schott blocking filter readings on a clear, sunny day were
compared. These readings agreed within the detection limit for the UV-MFRSR.
**Aerosol Optical Depth Retrievals**
After subtracting the large ozone and Rayleigh optical depth contributions to the total optical
depth, a residual remains that is assumed to be aerosol extinction. At Mauna Loa Observatory the
aerosol optical depths (AODs) are, in most cases, very small in the visible except in the
aftermath of volcanic eruptions (Dutton et al., 1994). The current paper examines AODs in the
ultraviolet near 305.6, 311.4, 317.5, 325.1, 332.4, and 367.8 nm where measurements of AOD
are infrequently made, especially below 340 nm. These wavelengths are shorter than those
measured by most sunphotometers with 340 nm the shortest wavelength measured by
AERONET (Holben et al., 2001), for example. Recently, however, López-Solano et al. (2018)
used Brewer spectrophotometers to derive AODs at five wavelengths between 306.3 and 320.1
nm. They compared AODs measured in this wavelength range by different co-located Brewers
and the UVPFR (Carlund et al., 2017).  In general, there was excellent agreement between the
Brewers and good, but less satisfactory, agreement between Brewers and the UVPFR, however,
there was no discussion of the wavelength dependence of the Brewer and UVPFR AODs at these
low wavelengths, which we consider next.
Figure 4 is typical of the AOD versus wavelength plots from the 43 days of measurements
plotted in Fig. 3. *Typical* visible wavelength dependent behavior indicates a negative slope on
this type of plot, however, the slope is positive from 305 to 332 nm and then becomes negative
after that, with the 368-nm wavelength AOD smaller than the 332-nm wavelength. The red point
in Fig. 4 is the average of co-located AERONET data at 340.8 nm (Holben et al., 2001) taken
during the same time as the average of the UV-MFRSR data plotted here. This plot indicates
consistency between the AERONET and UV-MFRSR data beyond 332 nm. A careful,
exhaustive analysis of uncertainties in the UVPFR paper by Carlund et al. (2017) that examines
this narrowband filter instrument at the shortest ultraviolet wavelengths close to those of our UV-
MFRSR could not explain the similar wavelength dependence (see the right-hand-side of their
Fig. 6) that they measured for low aerosol optical depth days in the autumn at Davos,
Switzerland. Their Figure 7 supports the argument that the Brewer spectrophotometer
measurements at similar wavelengths should return a similar wavelength dependence. However,
data from Davos in the spring did not show the downturn in AOD at the shortest wavelengths
that the autumn data indicated. In summary, Carlund et al. (2017) suggests that the size of the
uncertainties cannot completely rule out the possibility of a more typical wavelength dependence
with AOD increases with decreasing wavelength.
We looked at nitrogen dioxide ($NO_2$) as a possible contaminant that if not removed could explain
this wavelength behavior, however, the typical amount of $NO_2$ in the column above Mauna Loa
would necessitate a correction of less than 0.001 optical depths at 332 nm, less at the shorter
wavelengths, and slightly more at 368 nm. When only considering the 332 nm and 368 aerosol
optical depths the plot indicates the typical visible wavelength dependence. Although Fig. 4 is
the only plot of AOD shown, all of the 43 days had similar behavior.

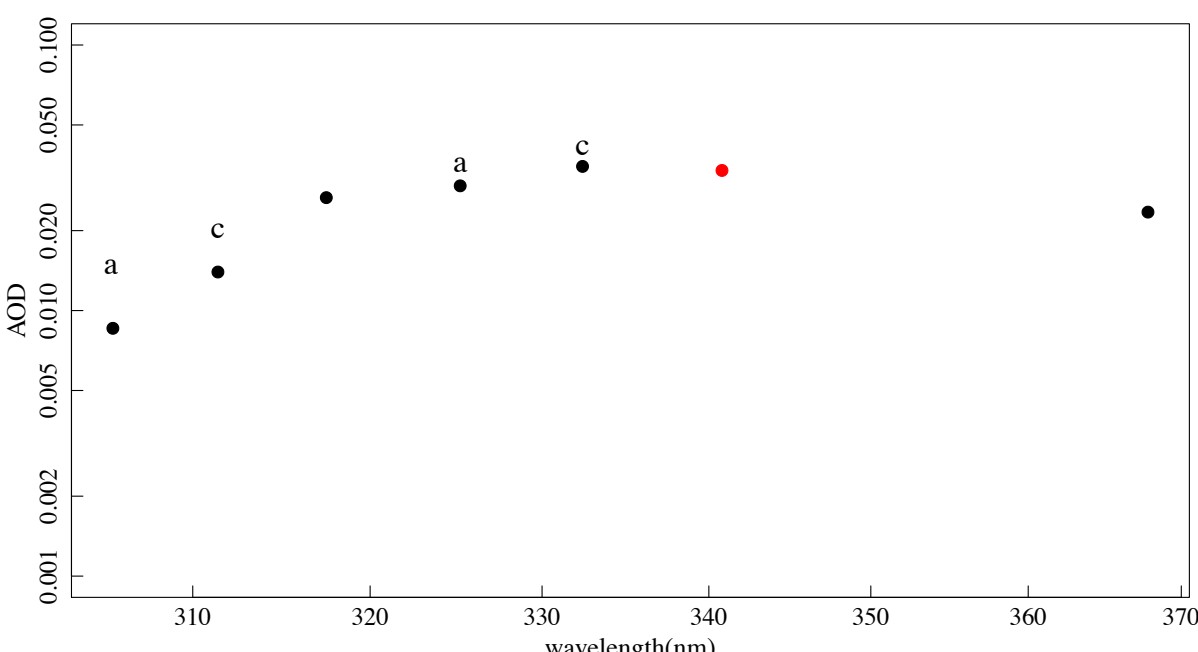

**Figure 4. This plot indicates the AOD versus wavelength for the UV-MFRSR filter set at Mauna Loa**
**Observatory. Instead of a negative slope, this figure, which is typical of the 43 days in this study, indicates a**
**positive slope with a negative slope indicated only by the two longest wavelengths. The 'a' and 'c' labels are**
**included to indicate the wavelength pairs used for the ozone retrievals. The red point is the average of the**
**AERONET points at 340.8 nm that overlap with the UV-MFRSR averaging period.**
**Discussion**

This paper focuses on data from the Manua Loa Observatory only. It corroborates results reported by Gao et al. (2001) regarding the UV-MFRSR's ability to retrieve ozone column that is in agreement with the Dobson instrument at Mauna Loa Observatory. Figure 3 demonstrates this agreement even though there was no attempt to synchronize ozone observations other than to have them occur on the same day.

Aerosol optical depths were measured in this very clean environment with expected low values, but an unexpected wavelength dependence. This wavelength dependence is similar to that obtained with an independent, sun-pointed narrowband filter instrument developed and operated at the World Radiation Center (WRC) in Davos, Switzerland. Our and the WRC's attempts to explain this wavelength dependence have yet to yield an understanding of the physics at work here. Systematic biases may be responsible; a better understanding of the very large optical depths associated with ozone absorption and Rayleigh scattering at these wavelengths that have to be subtracted to obtain the small AOD at these wavelengths may require more investigation. On the other hand, further study of environments with somewhat larger aerosol optical depths may indicate that this is, perhaps, associated with aerosol size distributions in some conditions.

**Appendix**

After the paper was accepted as a preprint in Atmospheric Measurement Techniques we were contacted by Alexander Smirnov of the AERONET team (aeronet.gsfc.nasa.gov). He made us aware of early Russian papers that measured AODs near the same short UV wavelengths that are plotted in Figure 4. These are discussed in a book by Rozenberg (1966) that was originally published in Russian in 1963, and translated to English for the 1966 publication in the reference list. Figure 97 in the Rozenberg (1996) book is a reproduction of the figure from the paper by Rodionov et al. (1942) that clearly shows AOD decreasing shortward of 380 nm (dubbed by these authors "anomalous transparency"). The observations were made at a high (3 km) mountain site explaining the low AOD values. These authors suggested that a specific aerosol size distribution might explain their wavelength dependence. Rodionov et al.'s (1942) measurements and suggested explanation of them were criticized, but a paper by Sakerin et al. (2000) suggesting that this effect and other unusual spectral dependencies of the AOD could be explained theoretically using specific combinations of nucleation, accumulation, and coarse aerosol modes.

*Author contributions:* JM drafted the paper and produced the figures. GM produced the data for Fig. 2 and provided details about the Dobson ozone retrievals using the AD and AC pairs.

*Competing interests.* The contact author has declared that none of the authors has any competing interests.

*Acknowledgments.* This paper benefited from Dobson ozone retrieval discussions with Peter Effertz and Irina Petropavlovskikh. Kathy Lantz provided the UV-MFRSR data from Mauna Loa Observatory and performed the out-of-band rejection studies. Gary Morris and Kathy Lantz provided a careful reading of the draft paper. Thomas Carlund provided useful insight on WRC's efforts at ultraviolet AOD retrievals using the World Radiation Center (WRC) UV-PFR while he was on sabbatical at the WRC in Davos, Switzerland. Alexander Smirnov was very helpful in

pointing out and discussing the earlier papers on Russian measurements and possible
explanations for the low UV short wavelength AODs.

*Financial support.* The publication costs for this paper were covered by the Global Monitoring
Laboratory of the National Oceanic and Atmospheric Administration.

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
