# Peer review of "Ozone and Aerosol Optical Depth Retrievals Using the Ultraviolet Multi-Filter Rotating Shadow-band Radiometer"

_Atmospheric Measurement Techniques, 2023_

## Author Response (AR1)

*Referee 1 Comments (*normal typeface*) and Replies(in italics):*

The paper is very interesting and shows the UV MFRSR instrument potential for total column ozone and aerosol optical depth (AOD) retrievals.

Concerning the introduction I think it would be important to introduce also the AOD retrievals and needs for AOD in the UVB region. There have been Brewer derived intercomparisons (e.g., Solano et al., 2018) but in general most known AOD retrieval networks do not go lower than 340 nm. AERONET, GAWPFR, SKYNET networks do not measure in the UVB so such studies are valuable.

*We have changed the paper to introduce and discuss ozone retrievals first, followed by an introduction and discussion of the aerosol optical depth retrievals, The Solano reference was incorporated into the AOD introduction.*

Instrument filters have a spectral responsivity so it can be commented that lambda in the equation is actually non monochromatic. This also impacts the calculation formulas as Rayleigh functions, ozone absorption have to be convoluted with filter spectral response function of each channel. Non accurate characterization of this spectral function for each of the filters could have an impact especially in AOD retrieval as the Rayleigh term is too big compared with the final retrieved AOD.

*The non-monochromatic nature of the retrievals is discussed explicitly in the first paragraph of 'Sources of Uncertainty' section.*

Probably a bit more discussion on how AOD related Langleys are taking into account possible intra-day ozone variations should be provided.

*The process used to choose acceptable Langleys (Michalsky et al., 2001) will eliminate Langleys that are influenced by large changes in ozone during a Langley plot. Further, there were only a few days where the standard deviation of the ozone sampled changed by more than 5 DU during any one day as is typical for this latitude.*

Since a CIMEL instrument measuring at 340nm is situated in Mauna Loa, it would be interesting to compare the CIMEL retrievals with and interpolated AOD at 340nm, based on the two highest UVMFRSR AOD retrieval wavelengths, for the 43 days of measurements. It will provide some additional assessment on the AOD retrieved levels. Interpolating at 340nm using 332 and 368nm based also on fig.3 should not be too uncertain.

*Figure 3 has been modified to show the CIMEL 340.8 nm AOD for the same averaging period as the UV-MFRSR points in this figure.*

The "negative Angstrom exponent" in the UVB range is really interesting but something that is difficult for me to understand and explain. Carlund et al., write: "As shown above, the uncertainties of the UV AOD values are, however, considerable and the AOD values measured

by the UVPFR are not significantly different from any of the extrapolated values in this low-turbidity case."   So more or less they say that they do not believe it.

*We would suggest that rather than "not believe it" (the low AODs at the shortest wavelengths) that they are suggesting that higher values more in line with a typical Angstrom exponent behavior are possible given the uncertainty in this measurement.*

It is interesting though why (if this is an uncertainty related aspect) that is systematic, always AOD being lower than what is expected.

*While not proof that the wavelength dependence is proven to be lower at the shortest ultraviolet wavelengths, there is some corroboration ii the Carlund et al. (2017) paper using data from the co-located Brewer (Carlund et al., 2017 Figure 7).*

Here, the authors more or less leave it like an open question, but in order to consider the fact that this can be real,  a further uncertainty analysis (that only the authors can do) should be done.

*The Carlund et al. (2017) uncertainty analysis was thorough, and we see no way to improve upon it.*

Langleys should work better or worse for different filters and the air mass factor dependence could lead to systematically low Vos (that is what is needed here to explain lower than usual AODs in the UVB). As already said filter functions and Rayleigh: depending on how broad they are and how well characterized could have an effect in AOD when multiplied by a Rayleigh based function in the UVB range. Air mass factors calculations are not mentioned. Langleys and intraday ozone ? direct sun calculations from global horizontal and diffuse irradiance from UV MFRSR (not so important though  for Mauna Loa) ?

*Air mass factors for Rayleigh and ozone are now addressed in the paper. Ozone variability during the day usually had a standard deviation of the sample that was much less than +/- 5 Dobson units. On occasion it was as much as +/- 7 Dobson units.*

Of course, all the above are some hints that without the data cannot be assessed properly.

In general, I think that the paper contributes a lot to the issue of ozone retrieval from the instrument and also for sure for the AOD at UVB wavelengths that has not been fully investigated mainly due to the  ozone related uncertainties in this spectral region.

López-Solano, J., Redondas, A., Carlund, T., Rodriguez-Franco, J. J., Diémoz, H., León-Luis, S. F., Hernández-Cruz, B., Guirado-Fuentes, C., Kouremeti, N., Gröbner, J., Kazadzis, S., Carreño, V., Berjón, A., Santana-Díaz, D., Rodríguez-Valido, M., De Bock, V., Moreta, J. R., Rimmer, J., Smedley, A. R. D., Boulkelia, L., Jepsen, N., Eriksen, P., Bais, A. F., Shirotov, V., Vilaplana, J. M., Wilson, K. M., and Karppinen, T.: Aerosol optical depth in the European Brewer Network, Atmos. Chem. Phys., 18, 3885-3902, https://doi.org/10.5194/acp-18-3885-2018, 2018.

*Reference added.*

*Referee 2 Comments (*normal typeface*) and Replies (italics):*

The paper verifies earlier research on the ability of the UV-MFRSR instrument to be used to retrieve total column ozone in good agreement with Dobson spectrophotometer. In addition to ozone retrievals results on aerosol optical depth determination down to UVB wavelengths using the UV-MFRSR is also presented. The authors report an unexpected wavelength dependence in AOD in the UVB, increasing AOD with increasing wavelength, under low aerosol conditions at a high-altitude site also found by others under similar conditions. Un-known systematic errors in the used ozone absorption and Rayleigh scattering coefficients might be one cause but it could also be connected to aerosol size distribution in some conditions. As the authors point out, this needs further research.

The paper is short and concise.

Questions and comments

For a more detailed description of the UV-MFRSR it is referenced to Gao et al. (2001). Especially, since the measurements are taken at UVB wavelengths it is important to know the bandwidth of the filter response functions. To make it clearer to the reader the authors should consider to include filter bandwidth values in the manuscript, e.g. around row 45.

*This information was added along with a figure and caption explaining these details.*

For a description of the calibration of the UV-MFRSR it is referred to Slusser et al. (2000) in the paper by Gao et al. (2001). It would help readers if the authors of the current paper directly refer to the paper by Slusser et al. as well as also briefly mention that the Langley results were corrected for the negative effect of finite bandwidths at the shortest wavelengths.

*We did not follow the Slusser calibration procedure so reference is not used. The wavelength assignments are not based on peak transmission, but on the integrated filter transmission above and below the assigned wavelength being equal. This could result in the wavelength assignments that are slightly shorter than they effectively are, but the magnitudes of the AODs should be correct.*

Reference: J. R. Slusser, J. Gibson, D. S. Bigelow, D. Kolinski, P. Disterhoft, K. Lantz, and A. Beaubien, "Langley method of calibrating UV filter radiometer," J. Geophys. Res. 105, 4841–4849 (2000).

*Reference was not added.*

In the total column ozone comparison between UV-MFRSR and Dobson it is stated that the UV-MFRSR uses daily medians of all 20 seconds samples taken at airmass less than three and which passed the cloud screening. How was this cloud screening done? And was it the same for both ozone and AOD retrievals?

*We used the consistency of the 20-sec data time-series to screen for clouds as explained in:*

*Michalsky, J., Denn, F., Flynn, C., Hodges, G., Kiedron, P. Koontz, A., Schlemmer, J., and Schwartz, S. E.: Climatology of aerosol optical depth in north-central Oklahoma: 1992:2008, J. Geophys. Res., D07203, https://doi.org/10.1029/2009JD012197, 2010.*

The uncertainties in the extraterrestrial responses at the wavelengths used in the ozone retrieval are very low, 0.2 – 0.3 %. This is very low, especially in the UVB. How were these very accurate results derived?

*The method we used to calculate uncertainty in Langley calibrations is explained in Michalsky et al. (2001). It comes from this methodology and the very stable conditions at Mauna Loa Observatory.*

*Note: An appendix was added to the paper because a reader of the preprint of this paper (Alexander Smirnov) made us aware of earlier Russian measurements that generally resemble the wavelength dependence that we found in Figure 4 of this paper, namely decreases in AOD at the short ultraviolet wavelengths. The AODs in that figure cited in the Appendix and reference list start to decrease at a slightly longer wavelength near 380 nm rather than 332 nm, but the tendency is the same.*